# Model Averaging and Augmented Inference for Stable Echocardiography Segmentation using 2D ConvNets

**Author(s) names withheld**                                          EMAIL(S) WITHHELD

## Abstract

The automatic segmentation of heart substructures in 2D echocardiography images is a goal common to both clinicians and researchers. Convolutional neural networks (CNNs) have recently shown the best average performance. However, on the rare occasions that a trained CNN fails, it can fail spectacularly. To mitigate these errors, in this work we develop and validate two easily implementable schemes for regularizing performance in 2D CNNs: model averaging and augmented inference. Model averaging involves training multiple instances of a CNN with data augmentation over a sampled training set. Augmented inference involves accumulating network output over augmentations of the test image. Using the recently released CAMUS echocardiography dataset, we show significant incremental improvement in outlier performance over the baseline model. These encouraging results must still be validated against independent clinical data.

**Keywords:** Convolutional Neural Networks, Echocardiography, Segmentation, Data Augmentation

## 1. Introduction

Echocardiography is a ubiquitous imaging modality for diagnosing and managing patients with cardiovascular disease (Virnig et al., 2014), a major cause of morbidity and mortality globally. Derived from the apical two- and four-chamber views (AP2/AP4) of an echo study, the left ventricular (LV) ejection fraction (EF) is the most common clinical index for measuring cardiac function. The time–consuming nature of the required manual delineations, and their high degree of inter-observer variability (Wood et al., 2014), has motivated the development of automatic techniques (Zhang et al., 2018).

Among many automatic segmentation methods that have been proposed in echo over decades (Noble and Boukerroui, 2006), convolutional neural networks (CNNs) have recently shown the most promise. In order to catalyze further development in this field, Leclerc et al. (2019) recently published the large annotated CAMUS dataset, providing expert manual annotations of hundreds of individual echo frames, needed for the supervised training of such models. The authors also tested numerous deep learning and prior segmentation techniques, reporting that deep CNNs produced the best results.

However, as pixel-wise classifiers without shape or topological constraints, typical CNNs can suffer from catastrophic failures, particularly in poor quality images or those with artifacts. While rare, these failures make CNNs not yet trustworthy for large-scale precision medicine applications using clinical data. To address such outliers Oktay et al. (2017) proposed an anatomically constrained CNN in 3D echo, where the training is regularized by

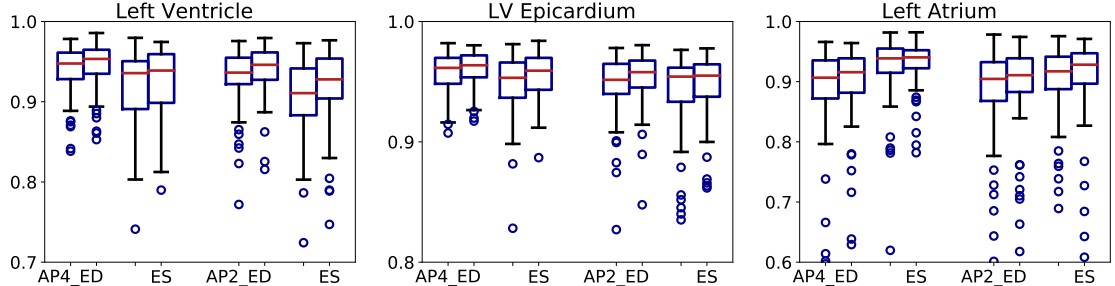

Figure 1: Dice distribution for each structure, by view and phase. The left side of each pair represents a single trained model; the right, the 8-fold model average.

an additional loss based on compact encoding of the ground-truth labeled images. However, Leclerc et al. (2019) could not reproduce those benefits on the 2D CAMUS dataset. Additionally, C.Qin et al. (2018) have integrated CNN-based segmentation with motion estimation in 3D cardiac magnetic resonance.

In this work we appropriate bootstrapping concepts to develop and validate two relatively practicable techniques for mitigating these outlier errors in 2D CNNs. The first is model averaging, in which a test image is segmented by multiple instances of a CNN trained with data augmentation over a sampled training set. The second technique is augmented inference, in which model output is accumulated over multiple augmentations of the test image. We use these techniques on the CAMUS dataset and show significant incremental improvement in outlier performance over the baseline model.

## 2. Methods

In this section we briefly describe our CNN model, data augmentation, evaluation, and experimental setup. Our model architecture is based on the popular U-net CNN (Ronneberger et al., 2015). With ∼13M parameters, the model uses convolutional down- and up-sampling, additive skip connections, and group normalization(Wu and He, 2018) for improved stability.

To help regularize output, we train all models with data augmentation reflecting the variability observed in the CAMUS set and echocardiography studies generally. The augmentations are performed on the fly and include random intensity windowing, rotation about the transducer, and additive Gaussian noise.

To evaluate performance, we report Dice overlap on the segmented 2D echo frames. For $S_{auto}$ and $S_{ref}$ representing the areas enclosed by the respective object contours, Dice overlap measures the intersection area divided by the average, $D(S_{auto}, S_{ref}) = 2(|S_{auto} \cap S_{ref}|)/(|S_{auto}| + |S_{ref}|)$. Dice is a highly validated measure in 2D.

The publicly-released CAMUS dataset consists of 450 patients, two (AP2/AP4) views per patient, and two annotated (diastolic/systolic, ED/ES) phases per view, totalling 1800 echo frames and corresponding label masks (background, LV endocardium $LV_{endo}$, LV epicardium $LV_{epi}$, and the left atrium LA). Additional information for each patient includes

age, sex, and reported ED/ES LV volumes and EF, along with the observed image quality for each view.

We initially generated ten patient folds, stratified on both EF range ($\leq 45\%, \geq 55\%, else$) and reported AP2 image quality (good, medium, poor), as suggested (Leclerc et al., 2019). We then excluded two folds for a test set totalling 90 patients (20%). We then performed 8-fold cross-validation training on the remaining patient folds: each iteration, the CNN is trained on seven folds while being validated against another for parameter optimization. Each view is trained separately, resulting in eight model instances per view that can generalize to the test patients.

## 3. Results

To evaluate model averaging, we compare the 8-fold accumulated inference to a baseline model of an arbitrarily chosen single fold. The box plots of Figure 1 clearly show that model averaging improves median performance and tightens the interquartile range across all structures, views, and phases, with a similar number of outliers ($[-3, +2]$ out of 90). To evaluate augmented inference, we consider an outlier of the baseline model in Figure 2. We accumulate the model inferences over 200 augmentations of the echo frame, as inference is relatively inexpensive. The recorded rotational augmentations are inverted before accumulation. As a result of augmented inference, Dice scores are dramatically improved over single inference for all labels ($LV_{endo}$ 0.69-0.83, $LV_{epi}$ 0.80-0.95, LA 0.36-0.70).

## 4. Conclusions

Model averaging and augmented inference are relatively practicable methods that can significantly mitigate catastrophic errors in 2D CNNs. Model averaging significantly reduces interquartile ranges, while augmented inference may dramatically improve segmentations of outlier test images, such as those with imaging artifacts. Future work revolves around incorporating video information and generalizing to other clinical datasets.

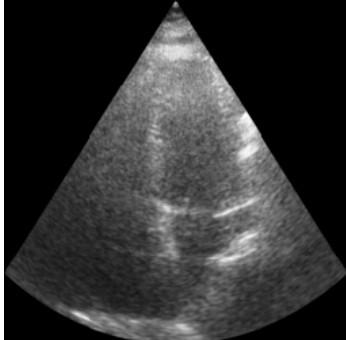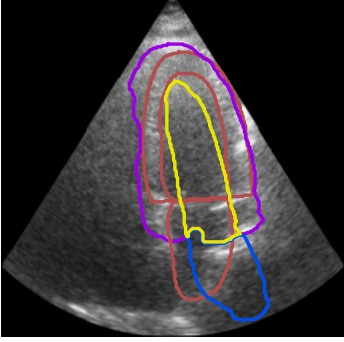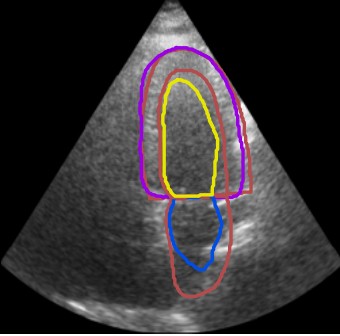

Figure 2: Augmented inference on a test case. Center frame: baseline model performance on test image ($LV_{endo}$ yellow, $LV_{epi}$ magenta, LA blue, ground truth red). Right frame: accumulated performance of augmented inferences of the same model.

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
