# OpenReview forum: "Model Averaging and Augmented Inference for Stable Echocardiography Segmentation using 2D ConvNets"
_MIDL.io/2020/Conference — Submitted to MIDL 2020_

### Official Review · AnonReviewer4 · 2020-03-14
**Interesting paper but rather incremental contribution with very brief experimental evaluation**

**Rating:** 2
**Confidence:** 4

**Review:**

The paper proposes to improve results of echocardiography imagery segmentation using model averaging and augmented inference. These ideas are not particularly novel, but have proven to be valuable in multiple recent studies.
In particular, the authors claim that averaging the predictions from multiple models improves performance and avoid the spectacular failures the single model prediction may sometimes exhibit. Additionally, data augmentation at test time also improves the results making them more stable.

The authors have trained and evaluated their method on data from the CAMUS dataset. This dataset is pretty large and the data variability observed there is sufficient to evaluate the generalisation capabilities of the method proposed by the authors.

Unfortunately, I find that the evaluation is not complete. First of all the authors only compare a randomly picked model from their 8-fold cross validation strategy with the average of the 8 fold. Would be interesting to see how a single model performs compared to an average of 2, 3, 4,..., 8 models. More importantly, it would be very good to see how the average of different architectures would work.

Additionally, the authors seem to state that test-time augmentation has been only done on one example, which is the one used for qualitative analysis and that is reported in figure. It would have been really great to see a formal comparison of the performance with and without test time augmentation for the whole test set.

Importantly, the box plot visualisation of the results leaves too much to the imagination of the readers. It would have been much better to include a table with results. Through a table, it would have been possible to show results for more experiments, even though some visibility on outliers might be lost (compared to box plots).

I have no doubt that the technique proposed in the paper is valuable. Given the length constraints of short papers I also understand the fact that the experimental evaluation is compact. I still think it could have been better, via a table and show different angles over the advantages brought by the proposed technique.

---

### Official Review · AnonReviewer3 · 2020-03-19
**Model for improved performance on outliers on echocardiography segmentation using test-time augmentation and ensembling**

**Rating:** 2
**Confidence:** 4

**Review:**

- The paper is very clearly written and the methods clearly described. The method involves ensembling 8 U-net models, trained on different overlapping folds of the echocardiography data with on-the-fly augmentations, and then applying test time augmentation by introducing 200 rotation variations and averaging the (unrotated) predictions.
- The data is split into 10 folds initially, where 2 are held out as test data. The 8 U-net models are trained on 7/8 remaining folds in rotation (with the remaining 1/8 held out for validation on each of these splits). The ensembled prediction is compared to a baseline U-net trained only on a single fold. This however is not a fair comparison, as the ensemble ultimately sees all the data from the 8 folds across the 8 trained models, so the baseline effectively learns from 12.5% fewer real training images.  Nonetheless, it is well established that ensembling improves over single models, as also demonstrated in the paper.
- Test time augmentation improves segmentation results compared to the baseline model too. It is unclear whether test time augmentation improves over the ensemble model without test time augmentation however.
- Both ensembling and test time augmentation are well established approaches in the literature. There is limited novelty in the proposed work, although clear improvements over a U-net baseline are shown.

---

### Official Review · AnonReviewer5 · 2020-03-20
**Lacking improvements and results**

**Rating:** 1
**Confidence:** 3

**Review:**

Contributions:

The authors propose to do ensemble learning (1) to further improve dice score as well as "accumulating" the predictions of a single model over test-time augmentation (2) to improve outlier performance . The data used came from the CAMUS dataset, and the model is a U-Net, the same architecture used in the original CAMUS paper.

Method:
For contribution 1, the authors split the patients into 10 folds, kept two as the testing set, and trained eight separate models on the remaining folds, keeping a different separate fold as validation set for all the models. A different model was trained for the two views available for each patient, totaling 16 models trained.

For contribution 2, the authors "accumulate" the predictions of a single model trained over a single fold by augmenting a test image 200 times via a combination of intensity modification, rotation and Gaussian noise.

Results:
For contribution 1, a box plot of dice distribution is reported over different structures, separated by view and phase. The results are shown for a single model against the proposed ensemble of models.
For contribution 2, the dice score improvement for the accumulated result is reported for a single test image, as well as a qualitative assessment of the segmentation for the same image.

Criticism:
Ensemble learning is generally recognized as an easy way to improve results on virtually any task. However, it is not a cheap method and requires n-times the amount of memory and training time. In itself, the reviewer feels it cannot be considered an improvement of a method. In this particular case, as figure 1 shows, the ensembling can hardly be justified as improvements shown via box plot seem to be marginal, at a cost of 8 times the amount of memory.

Contribution 2 seems to have significant improvements over the baseline, the authors' own U-Net trained on a single fold. However, test-time augmentation is another commonly used practice and the reviewer also feels it is not a novel idea in itself. Furthermore, it is unclear what "accumulating" means, whether it is taking the overlap of the 200 predictions of the noisy image, a threshold per pixel, or any other method. Finally, the reported results are vague and only from a single hand-picked outlier test image. Nothing can be confidently inferred from this result.

While the two contributions are orthogonal, no results are reported on the application of the two contributions at the same time.

Finally, only the dice score is reported, while the original CAMUS paper also reported Hausdorff distance and mean absolute distance.

Conclusion:
The paper does not present any novel idea for cardiac segmentation. Even though the presented article is a short paper, the article glosses over important details and fails to present meaningful results.

---

### Official Review · AnonReviewer6 · 2020-03-20
**Review of paper: Model Averaging and Augmented Inference for Stable Echocardiography Segmentation using 2D ConvNets**

**Rating:** 2
**Confidence:** 4

**Review:**

-In the methods section, the authors claim to describe their model. However, all we have is the description of a U-Net. Any modification was made to the U-Net?
 - In the results section, it is not clear why the authors chose only one fold? Furthermore, it is unclear by how much the results got improved by the proposed method
 - In the conclusion it is said '... augmented inference *may* dramatically improve...', does that mean that it sometimes work and sometimes not? Please be more clear.

---

### Meta-Review · Area_Chair1 · 2020-03-27
**MetaReview of Paper302 by AreaChair1**

**Rating:** 2

**Metareview:**

This is a well written paper.  But like the reviewers, I lean towards a weak reject as the improvements of the proposed method are quite humble to say the least (c.f. Fig1).  Furthermore, clear statistics on the reduction of the number of outliers are missing.  This is too bad considering that this was the gaol mentioned in the abstract :

" However, on the rare occasions that a trained CNN fails, it can fail spectacularly. To mitigate these errors, in this work we develop and validate two easily implementable schemes for regularizing performance in 2D CNNs"

**Paper Type:**

validation/application paper

---

### Decision · Program_Chairs · 2020-04-11

Reject